# The Concept of “Cancer Stem Cells” in the Context of Classic Carcinogenesis Hypotheses and Experimental Findings

**DOI:** 10.3390/life11121308

**Published:** 2021-11-27

**Authors:** James E. Trosko

**Affiliations:** Department of Pediatrics/Human Development, College of Human Medicine, Michigan State University, East Lansing, MI 048864, USA; trosko@msu.edu

**Keywords:** cancer stem, cells, cancer initiating cells, Oct4 gene, gap junctional intercellular communication, multi-stage, multi-mechanism hypothesis of carcinogenesis, stem cell hypothesis of carcinogenesis, de-differentiation hypothesis of carcinogenesis

## Abstract

In this *Commentary*, the operational definition of *cancer stem cells* or *cancer initiating cells* includes the ability of certain cells, found in a heterogeneous mixture of cells within a tumor, which are able to sustain growth of that tumor. However, that concept of *cancer stem cells* does not resolve the age-old controversy of two opposing hypotheses of the origin of the cancer, namely the *stem cell* hypothesis versus the *de-differentiation* or *re-programming* hypothesis. Moreover, this *cancer stem* concept has to take into account classic experimental observations, techniques, and concepts, such as the multi-stage, multi-mechanism process of carcinogenesis; roles of mutagenic, cytotoxic and epigenetic mechanisms; the important differences between ***errors of DNA repair*** and ***errors of DNA replication*** in forming mutations; biomarkers of known characteristics of normal adult organ-specific stem cells and of cancer stem cells; and the characteristics of epigenetic mechanisms involved in the carcinogenic process. In addition, vague and misleading terms, such as *carcinogens, immortal and normal cells* have to be clarified in the context of current scientific facts. The ultimate integration of all of these historic factors to provide a current understanding of the origin and characteristics of a *cancer stem cell*, which is required for a rational strategy for prevention and therapy for cancer, does not follow a linear path. Lastly, it will be speculated that there exists evidence of two distinct types of *cancer stem cells*, one that has its origin in an organ-specific adult stem cell that is ‘initiated’ in the stem cell stage, expressing the Oct4A gene and not expressing any connexin gene or having functional gap junctional intercellular communication (GJIC). The other *cancer stem cell* is derived from a stem cell that is initiated early after the Oct4A gene is suppressed and the connexin gene is expressed, which starts early differentiation, but it is blocked from terminal differentiation.

“*The biochemistry of cancer is a problem that obligates the investigator to combine the reductionalistic approaches of the molecular biologists with the holistic requirements of hierarchies within the organism. The cancer problem is not merely a cell problem, **it is a problem of cell interactions, not only within tissues but also with distal cells in other tissues*****” [1]**.*“Some would argue that the search for the origin and treatment of this disease will continue over the next quarter century in much the same manner as it already has in the recent past, by adding further layers of complexity to a scientific literature that is already complex beyond measure. But we anticipate otherwise: those researching the cancer [or any other human disease] problem will be practicing a dramatically different type of science than we have experience over the last 25 years. Surely much of this change will be apparent on the technical level. But ultimately **the more fundamental change will be conceptual”*** [2].*“Personalized medicine is the latest promise of a gene-centered biomedicine to provide custom-tailored to the specific needs of patients. Although surrounded by much hype, **personalized medicine lacks the empirical and theoretical foundations necessary to render it a long-term perspective**. In particular, the role of genetic data and the relationship between causal understanding, prediction, prevention and treatment of a disease need clarifying”* [3].

## 1. Introduction: How Some Historic Experimental Findings and Hypotheses of Cancer Shaped Today’s Concept of Cancer Stem Cells

These three introductory quotes embody much of my concern in this *Commentary* as it concerns the concept of *cancer stem cells*. They span decades of cancer research by different disciplinarians, involving years of experimental research and philosophical reflection of their experiences of their field.

In this *Commentary*, I rely on those giants of cancer research and my own research experience of 50 years, to try to make sense of both great discoveries and ideas, as well as confusing and often contradictory uses of terms and concepts. With that as a framework of what is to follow in a non-linear historical fashion, I will try to use both experimental findings and concepts, as well as my own historic and philosophical musings, to generate a view of *cancer stem cells* that is testable.

Because it has long been known that cells derived from a patient’s cancer could outlast the patient, from whom they were derived, and could be perpetuated either in vitro or in experimental animals, it was assumed to have developed the property of immortality during the initiation of the carcinogenic process. Take for example, the HeLa cell line, derived from Henrietta Lang, which has been studied in laboratories all over the world and has been subject of thousands of research papers [4]. Consequently, one objective of many studies was to induce immortality in *normal* cells, in vitro, in order to determine the mechanism by which this happens and to use this protocol to determine if any physical, chemical or biological agent might be a *carcinogen*. Although there was some initial success using rodent cells for this purpose [5], trying to immortalize normal human cells met with failure [6,7,8]. When the new concept of oncogenes was introduced, genetically-engineering normal rodent cells with specific DNA sequences led the way to get insights on the *immortalizing* normal human cells [9,10,11].

However, before the next breakthrough that seemed to provide another view of the *immortalizing* process, the concept of *cancer stem cells* has to be viewed from the perspective of many classical experimental animal cancer studies and those from epidemiology. In many of today’s studies *of cancer stem cells*, these classic studies have been largely ignored. When Percivall Potts correlated the unusually high frequency of scrotal cancers in chimney sweepers and the soot from the combustion of fuel [12], a link was formed to chemicals in the soot that seemed to be *carcinogen.* One school of thought was that these chemicals must have an irreversible effect on the genome (DNA) and later studies did show some of these chemicals could attack the DNA [13]. However, at that time, animal experiments [14,15], in general, seemed to indicate that the carcinogenic process was not a one-hit process, by which a single normal cell could be irreversibly altered to become an invasive, metastatic cancer cell [16,17,18]. The new concept emerged suggesting that carcinogenesis was a multi-step, multi-mechanism process, consisting of an **initiation** of a single “normal” cell to become “immortal”, followed by a **promotion** event over a long period of regular exposures at a threshold level to clonally amplify this single “initiated” cell into a benign tumor, which then transitioned to become an invasive and metastatic cell by the **progression** process.

Another important concept that added to this classic understanding of this **initiation/promotion/progression** process was that, even though all the cells in the tumor appeared to be heterogeneous in terms of their genotypes and phenotypes, they were derived from a single common “normal” ancestor [19,20].

Added to this new concept were two opposing hypotheses as to which single cell gave rise to these cancer cells. The *stem cell* hypothesis [21,22,23,24,25,26,27,28], and the *de-differentiation* or *re-programming* hypothesis [29] emerged. The late Dr. Van R. Potter conceptualized the stem cell hypothesis as: “Oncogeny as partially-blocked ontogeny” [1].

## 2. The Stem Cell versus the De-Differentiation Hypotheses: The Origin of the “Cancer Stem Cell”

Now back to how the concept of *cancer stem cells* emerged and how the current controversy of the origin of the cancer stem cells seems to be unresolved. In the history of science, the intellectual journey from the starting idea of any explanation for a disease causation to the current hypothesis is never a linear journey. Ideas that ultimately get to the current state come by jumps, starts, rejection or refinement of disproven paradigms, parallel experimental, methodological or conceptual advances in different disciplines, and of course, serendipity. To finally merge all these disconnected ideas, it takes what was once said: “*Research is to see what everyone has seen and think what nobody has thought”* [30]. To make clear, several words have been used in the history of trying to understand the mechanisms of carcinogenesis, namely, *normal cells, carcinogens and immortal*, are now seen as very misleading. As will be shown later, what is the *normal* cell that is being converted to ultimately become a *cancer stem cell*? What is a *carcinogen,* a concept that implies an agent that can induce all three phases of the multi-step, multi-mechanism process of carcinogenesis, even after a single exposure? Last, is a cell being classified as *immortal* due to an induced change in the phenotype caused by some irreversible process, such as a mutation, or by this mutation causing the cell to stay in its natural immortal state. These are important issues that need to be resolved.

While this brief review of the history of carcinogenesis is not linear in time, in retrospect, the early events now make sense in view of recent characterizations of stem cells. Recall the early concept that stem cells might be the target cells for the start of the carcinogenic process; when this concept was developed, no one had even isolated or characterized a real stem cell. The idea of stem cells in developmental biology and embryology was a reasonable logical concept. After it was shown that in vitro experimental approaches, using soft agar growth, might help us to understand the mechanisms of carcinogenesis, the induction of apparent tumors that appeared when *normal* cells were exposed to some agent (physical, chemical, biological). To test if these abnormal looking clones of cells are really tumorigenic, they were placed in soft agar or injected back into an immune-deficient rodent. If tumors appeared after this procedure, the agent that brought about the in vitro and in vivo tumors was assumed to be a *carcinogen* and probably a mutagen. However, if only a few cells of the soft agar clones were used to be injected in the immune-deficient rodent, no tumors were seen in vivo, unless large numbers were injected. At the time of these early experiments, no one had any idea why this was the case. In addition, several in vitro assays to determine if an agent that contributed to this conversion of a “normal” cell to one that gave rise to a tumor in an immune-deficient rodent, it was assumed to be a mutagen or genotoxicant. The Ames assay and many other so-called mutation assays were shown to be non-consistent with one another for a number of reasons. Even later, when real mutations at the molecular level were found, both in vitro and in vivo, in the cells shown to be tumorigenic, problems of interpretation again arose, since the DNA studies were not determined to be either the genomic DNA or mitochondrial DNA or that the methods of isolating the DNA might have contributed to the measurements. In brief, major challenges to the interpretation of these mutation assays have been discussed [31,32].

One early observation was that of discrepancies in these in vitro assays to find transformed cells, few of which actually cite this paper; a study was designed to determine why, from lab to lab, or day to day in the same lab using identical chemicals and cells, as well as protocols, dramatically different results were found [33]. To make this long story short, it was shown, using a pool of Syrian embryo cells to test the presumptive chemical carcinogen, that if the population of *normal* cells had a few cells that seemed to have no contact-inhibition, one could obtain transformed cells after the application of the presumptive *carcinogen*. If the population from this pool of Syrian hamster embryo cells did not have in its population the type of no-contact inhibited cells, no amount of the presumptive *carcinogen* would induce transformed cells. The clue was that only the few cells that did not have *contact inhibition* gave rise to transformed cells. The clue was *contact inhibition* [34].

In another disciplinary field, the work of Werner Loewenstein and Kanno [35], as well as the freeze fracture pioneer Dr. J.P. Revel [36], fused the fields of electrophysiology and electron microscopy to identify a structure of cell membranes (gap junctions) to a physiological function of this structure to synchronizing both metabolic and electrotonic functions of cells in tissues. Later, Borek and Sachs [37] and Borek et al. [38] noted that *normal* cells, which had gap junctions, could *contact inhibit* or have growth control and differentiate, as well as have the potential to become senescent. On the other hand, cancer cells that do not *contact-inhibit* or have growth control, cannot terminally differentiate, but were *immortal*, and also had no functional gap junctional intercellular communication [39].

The terms, *senescent*, *immortal* and *normal* cells appear, again and again, in the cancer literature. However, it has now been shown that *normal* primary human fibroblasts cells would, through replicative replication, senesce after about 50 cell passages in vitro [40]. However, later it was shown that, given the manner by which human primary biopsies that gave rise to these fibroblasts were grown at ambient oxygen levels, they followed Hayflick’s observation. Yet, if these primary fibroblasts were grown at very low oxygen levels, they could be passaged much longer [41,42,43,44,45]. It was as though oxygen was a toxic agent to some cells in the population that were needed for their sustained growth. Even later, it was shown in our laboratory that early passages of skin fibroblasts contained adult stem cells [46,47]. Our results, which are currently unpublished, have shown that very early primary human fibroblasts express a key stem cell marker, Oct4A. Could oxygen levels affect the stem cell state of stemness?

## 3. Clues to Prove the Stem Cell versus De-Differentiation Hypotheses of the Origin of Cancer Stem Cells

Having these published observations in several disciplines in mind, in the context of my laboratory wanting to resolve the issue as to whether the *Stem Cell* hypothesis or the *De-differentiation* hypothesis was correct as being the origin of the “cancer stem cell” hypothesis, the question was: “*How could a normal stem cell in any tissue/organ survive without being forced to differentiate by close differentiated offspring that had functional gap junctions?”* Only two possible explanations seemed reasonable. First, the stem cells were sequestered by some physical barrier that prevents contact with these gap junction-expressing cells, or second, these stem cells did not express their connexin, or gap junction, genes. We then designed what we called the **kiss of death** assay [48].

This assay was based on disassociating all the cells of a normal human organ biopsy, which contained three kinds of cells, namely, a few rare organ-specific adult stem cells, the large numbers of progenitor cells and the terminally differentiated cells. We assumed that the stem cells did not have either expressed connexin genes or have any functional gap junctions. The progenitor cells would have functional gap junctions, while the terminally differentiated cells might or might not have gap junctions, but they could not divide. Next, with approximately a million of these disassociated cells, they were placed on a confluent mat of normal human fibroblast, which were lethally irradiated and were unable to proliferate. Once the progenitor cells attached to the confluent mat of fibroblasts, they formed gap junctions with the proliferative- disabled fibroblasts and eventually died. The terminally differentiated cells never formed any clones on the mat. Since they either died by apoptosis or because they could not proliferate. On the other hand, after a week, a few small clones of cells appeared to be proliferating. After these clones were removed, they were tested for expressed connexins and functional gap junctions. None were found. These cells were then exposed to various differentiating agents, and then they expressed connexin genes, had functional gap junctions and ultimately differentiated (see Figure 3 in [48]).

Later, when it was shown that one of the biomarker genes of embryonic stem cells was the Oct4 gene [49,50,51], our lab had several organ-specific adult stem cells (kidney, breast, pancreas, and later, liver). We tested them for expressed Oct4 and non-expressed connexin genes [52]. This confirmed our hypothesis that the clones we isolated via **the kiss of death** assay and later other techniques [53] were, in fact, true adult stem cells. We decided that only two of our 20,000+ genes needed to be tested as biomarkers for any stem cells. Oct4 is required for maintaining stemness and the connexin genes and functional gap junctions are required for differentiation, growth control, apoptosis [54] and senescence.

## 4. A Test for Stem Cell and De-Differentiation Hypotheses for the Origin of the “Cancer Stem Cells”

Even though speculations and experimental tests were reported to support the stem cell hypothesis [22,23,24,25,26,27], no reports used a single isolated human adult stem cell to put these hypotheses to a test. Using normal human adult stem cells as the target cell of the initiation/promotion/progression carcinogenic in vitro process, we tested these cells for the expression of Oct4A and for functional gap junctional intercellular communication [23]. Oct4A was expressed but no connexin43 was expressed and there was no functional gap junctions in the human breast stem cells. We then tested whether these human breast adult stem cells could be differentiated into breast epithelial cells and whether Oct4A was still expressed and whether the connexin43 was expressed, and if they had functional gap junctions. These differentiated breast epithelial cells had no expressed Oct4A, but did express connexin43 and had functional gap junctions (see Figure 1 in [23]).

Next, the normal human adult breast stem cells and the differentiated breast epithelial cells were transfected with the large T gene of the SV-40 virus; only a few clones of proliferating breast stem cells were obtained. These cells still expressed the Oct4A gene and they had no functional gap junctions. These cells were apparently *immortal*, but not tumorigenic when tested in immune-suppressed mice. No *immortalized* cells were derived from those differentiated breast epithelial cells, confirming what many previous studies had shown, that to *immortalize* normal differentiated cells was either difficult or impossible [6,7,8].

The *immortalized* human breast stem cells were now X ray-irradiated, and a few clones that formed soft agar clones were isolated. These were tested for tumorigenicity in immune-deficient mice and they formed slowly or weakly growing tumors. Next, these X-irradiated cloned cells were treated with the ErB2/Neu gene and several clones that had significant rapid growth in soft agar were tested for tumorigenicity in immune-suppressed mice. In this case, the tumors were very tumorigenic and still expressed Oct4A gene and did not have functional gap junctional intercellular communication.

The take-home message of this experiment on a clonally-derived series of adult human breast stem cells showed that the tumorigenic breast cell line was directly derived from the normal adult stem cell that expressed the Oct4A gene and did not have functional gap junctions. In other words, the biomarker gene, Oct4A, was not induced by the carcinogenic process, but remained expressed from the start of the initiating event. Moreover, the initiating event blocked the differentiation process, as the late Dr. Potter predicted (“Oncogeny as partially blocked ontogeny” [1]). In addition, since the original stem cells are naturally “immortal” until they are terminally differentiated or become *mortal,* the so-called *immortalizing* viruses, e.g., SV40, are not immortalizing a normal *mortal* cell, they are blocking *mortalization*. These types of immortalizing viruses should be re-named. In other words, this experiment adds to those speculated hypotheses and actual direct experiments that strongly suggest the *stem cell* hypothesis is the correct hypothesis for the origin of cancers.

The recent demonstration of the isolation of *induced pluripotent stem cells* has given renewed support for *re-programming or the de-differentiation* of normal differentiated somatic cells to become “immortal”, or embryonic-like [55]. While no one can doubt that genetically-engineering a population of normal differentiated somatic cells with the Yamanaka embryonic genes, includingOct4, cannot produce “iPS” cells. However, there is now a legitimate reason to challenge *the interpretation that re-programming took place.* Clearly, that original population contained a few fibroblast stem cells [46,47]. These few adult fibroblast stem cells that, when transfected with these embryonic genes, now had both their own endogenous Oct4 gene expressed, but also those of the exogenous Oct4 that were introduced in its genome. That gave these few fibroblast adult stem cells the growth advantage over the somatic differentiated non-Oct4 expressing cells. Therefore, those using “iPS” cells for all kinds of experiments are really using normal fibroblast stem cells with their genome altered by the exogenous embryonic genes. In effect, these “iPS” cells are not really the result of “re-programming”, but rather the selection of pre-existing adult stem cells of the original primary culture of human tissue.

Now, one has to demonstrate if these tumorigenic cancer cells that were derived from a single normal immortal adult specific stem cell contain the *cancer stem cells.*

## 5. Characteristics of the Cancer Stem Cells

Today, these terms, *cancer stem cell* or *cancer-initiating cell*, are defined, operationally, as the cell that has the ability to sustain the long-term growth of a tumor, having all the characteristics of a tumor from which it was derived. One of the first clues came from a creative experiment, using Hoechst dye to stain cells of a tumor [56]. When these cells were placed in a cell sorter, two populations of cells were obtained, one fluorescing and a small population that did not incorporate the dye, hence not fluorescing. These latter population were classified as ***side population*** cells. These ***side population*** cells were shown to develop into tumors, manifesting the same characteristics as the tumor from which they were derived. Hence this procedure to isolate *cancer-initiating* or *cancer stem cells* became the operational definition of the *cancer stem cells*. These cells did not retain the Hoechst dye because they expressed functional drug transporter genes. Therefore, the Hoechst dye-containing cells or the *cancer non-stem cells* did not express those genes; hence, the dye, which binds to DNA, indicated that toxic chemicals could enter these types of cells and be killed. On the other hand, the ***side population*** cells would be resistant to potential toxic chemicals. Hence, this is a significant clue as to why most anti-cancer drugs, designed to kill cancer cells, can only kill *the cancer non-stem cells* but not the *cancer stem cells*.

At the time, one had to view this finding from an evolutionary vantage point. If during early evolution of the multi-cellular organism, when the stem cell appeared, that organism, with its few stem cells and many progenitor and terminally differentiated cells, if they were exposed to some toxic agent, and all cell types were equally susceptible to the toxic agent, the organism would not survive. The stem cells are needed for growth wound repair, tissue damage and the natural attrition of dying cells. Evolution selected for the stem cells to be able to be maintained in a low oxygen micro-environment, have various anti-oxidant systems to protect against free radicals, and to have both a nuclear membrane to act as another barrier to free radical production of the mitochondria of its neighboring differentiated daughters [57,58] and DNA repair enzymes [59].

## 6. Are All “Cancer Stem Cells” Identical?

The *cancer stem cell* of the breast, colon, liver, pancreas, etc. would have specific organ-specific markers. However, do they share some common markers that make them a *cancer stem cell*? As pointed out previously, if the cancer stem cell is derived from a normal organ-specific adult stem cell, and if one marker, namely, the Oct4A gene, is shared by all *cancer stem cells*, as seems to be indicated by many published papers [60,61,62,63,64,65,66,67,68,69,70,71,72,73,74,75,76,77], then it would seem that some strategy should be developed to find ways to either shut down this gene in order to force the cells to differentiate or apoptose. In addition, if the connexin genes and functional gap junctions are not found in either the normal stem cells or *cancer stem cells*, agents that might induce the connexin genes to be expressed could induce either differentiation or apoptosis. One such example of this happening has been demonstrated [78].

An additional series of observations have, however, complicated this simple solution. In a series of many canine tumor types, Oct4 was detected in over 95% of these tumors [79]. Yet, there were a few canine tumors where no Oct4 marker was detected in the tumor cell population. Again, to complicate the situation, virtually all tumors are found in two states, namely, the embryonic-like phenotype, or in the quasi-differentiated phenotype (basal cell skin carcinoma or squamous skin carcinoma; polyp-type colon carcinoma and the *flat-type* or embryonic-type colon carcinoma; lung small cell carcinoma or the non-small cell carcinoma, etc.). To my knowledge, no one has examined the *cancer stem cell* of either type of tumors of any organ to determine if the Oct4A is not expressed, but some connexin gene is expressed in the partially differentiated stem cell, and if *cancer stem cells* of the embryonic-type organ tumor have the Oct4A gene expressed and no connexin gene or functional gap junctional intercellular communication is found. If that prediction is confirmed, then only these two genes (possibly the drug transporter genes) need to be monitored.

In a recent paper, it has been predicted, based on many arguments presented here, that there exist two *cancer stem cells* [80,81]. Therefore, using any cancer therapy protocol against one type will not work against the other type in the same organ. Currently, there is no explanation as to why some *cancer stem cells* do not express the Oct4A gene, yet are still derived from the normal organ-specific adult stem cell, as are the Oct4A expressing *cancer stem cells*. One potential explanation is that, as an adult stem cell starts to differentiate, it starts to turn off the expression of Oct4A; it is initiated in this transition state. The connexin gene is expressed and it starts to differentiate, but does not achieve the ability to terminally differentiate (i.e., asymmetrical cell division is inhibited, but the symmetrical cell division is still functioning). In these cases, since it was shown that gap junction can be inhibited if some oncogene is also expressed, thus blocking the gap junction function. Only future experiments to test this hypothesis will affirm or deny its validity.

Last, one of the new insights that has emerged in the search for ways to target the cancer stem cells in a cancer patient is to find ways to minimize the unintended consequences of the toxic effects of the therapies being used. One of the side effects of killing cancer cells by radiation or chemotherapeutic cytotoxicants is that it triggers a cytokine storm [82,83], which is a natural response for an organism to repair tissue or the loss of dying cells. These various cytokines have been shown to modulate gap junctional intercellular communication [84].

Since the drug metformin has already been shown to protect against chemicals that can inhibit gap junctional intercellular communication to cause enhanced cell proliferation [85], and since metformin has already been shown to target *cancer stem cells* in a three dimensional human breast organoid [86], we predicted that using it together with any anti-cancer therapy to minimize both the side consequences of the therapy on the non-cancer cells, as well as to help sensitize targeting the cancer stem cells, would seem to be a rational strategy. Several reviews of the literature seems to have provided mixed results, with some studies showing no effects, others some negative effects, and some demonstrating positive effects [87,88,89,90]. Since metformin seems to act in a similar, but not identical, way to other chemicals that protect cells from agents that can inhibit gap junction intercellular communication, such as resveratrol, CAPE, green tea components, licorice components, caffeine, lycopene [91], as well as lovastatin [92], melatonin [86] and others, understanding all the different mechanisms by which agents can modulate gap junctional intercellular communication is critical. Phorbol esters, a powerful gap junction inhibitor, work by a very different biochemical cellular mechanism than does DDT. There are both receptor- and receptor-independent mechanisms by which inhibitors of gap junction function, for example low dose estrogen verses high dose estrogen, have different effects of triggering intracellular signaling. The mixed clinical trials might simply be due to not understanding the specific factors involved, namely, the individual’s genetics, gender, development stage, the specific intra-cellular signaling pathways that are triggered by the agents being used, dose to be used, timing with the anti-cancer therapy, knowing which of the two types of *cancer stem cells* is in the patient’s tumors, as well as the time of day the therapy is administered. If ever there was a support for precision medicine or personalized medicine, this could be an example. This is seen in the context of the definition of personalized or precision medicine which *refers to a medical approach in which diseases are diagnosed, prevented, and treated according to the context of each patient’s unique genetics, history, and lifestyle. Treatments are optimized and side effects are reduced, and this drastically reduces the overall cost of healthcare to society.*


## 7. The Ultimate Problem of Designing an Anti-Cancer Agent That Targets These *Cancer Stem Cells*

Strategically, one would like to prevent any future cancer over the treatment of an existing tumor or the wide-spread metastatic cancer. However, there are many obstacles that have to be overcome. The first is the problem of the initiation of a single normal cell (in this case, it is assumed that a normal cell is an adult organ-specific adult stem cell), because initiation, while preventable to a degree, it is not possible to eliminate all initiating events or a mutation in a critical cancer-associated gene responsible for blocking asymmetrical cell division of the organ-specific stem cell. One needs to not be exposed to too much UV light from the sun or to not sit on a uranium pile. However, mutations can be produced not only by the genomic DNA being damaged (***errors in DNA repair***) [93,94,95,96,97], but also by ***errors in DNA replication*** [98]. Consequently, even in the absence of genomic DNA damage, every time a stem cell replicates during a growth spurt, hormone growth factor, or cytokine stimulation during wound healing, cell death of a tissue and tissue removal, there is a finite probability that a “spontaneous” mutation in one of those *initiating genes* would occur.

Probably one of the most convincing proofs that a specific oncogene mutation in a tumor associated with lung cancer is the demonstration that mutations in non-smokers’ lung cancer cell’s Ha-ras oncogene were identical to the mutation in that gene found in lung tumor cells of smokers [99]. As an important side observation, it was a classic chemical found in cigarette smoke that was shown not to be a genotoxicant or mutagen, but rather an agent that increased the transformation frequency of baby hamster embryonic cells [100,101,102]. Later, studies on any of the predominant aromatic hydrocarbons in cigarette smoke showed them to be tumor promoters, but not initiators [103,104]. When using assays to detect agents that could reversibly inhibit gap junctional intercellular communication and shown that they were epigenetic-acting chemicals, the most predominate aromatic hydrocarbon was a non-mutagenic, 1-methyl anthracene [105]. Therefore, tying this set of observations together, it seems that lung cancers of non-smokers and smokers might be the result of a spontaneous mutation, caused by an error of DNA replication in a gene that blocks asymmetric cell division of a stem cell of the lung, which was promoted by endogenous or exogenous epigenetic chemicals.

Returning to the major problem of killing the *cancer stem cells*, how can an agent be designed to be given in vivo to target the *cancer stem cell* of a benign or malignant cancer? It is now accepted that all tumors are a heterogeneous mixture of non-cancer stem cells [106] and a few cancer stem cells, together with normal stromal cells and invasive immune cells. In general, there is a very wide range of genotypes (both chromosomal and gene-wise) in the cancer non-stem cells. To date, there is no solid evidence that the deviation chromosomal/gene mutations and chromosome instability, let alone epigenetic deviations of the initiated cells during its evolution, were responsible for the origin of the *cancer stem cell.*

First, as was pointed out before, we will never eliminate the origin of the initiated adult stem cell because it is usually surrounded by, and communicating with, its normal differentiated progenitor or differentiated daughters. Under those conditions, these communicating normal cells are sending signals to induce a normal phenotype in the initiated cell, in spite of having a critical mutation in a cancer-associated gene. This allows the “initiated cell” to escape the immune system, as it appears to be “normal” or self-like.

However, when these cell–cell communication-coupled cells are exposed to agents and conditions that can inhibit this communication process, these initiated cells can clone multiply to form those benign lesions, such as a papilloma of the skin, enzyme altered foci of the liver, polyp of the colon or nodule in the breast. Not all of these lesions will go on to form an invasive and metastatic tumor. In fact, some might even be blocked from further development or even regress [107]. However, a few of these lesions will have a cell within the benign lesions that will have acquired the “hallmarks of cancer” [2,108].

Faced with the problem of targeting any agent that might induce differentiation, apoptosis or cytotoxicity in vivo of any organ-specific tumor, one has to deal with a tumor that is a heterogeneous mixture of *cancer non-stem cells, cancer stem cells* and various normal stromal and immune cells. Moreover, a chemical anti-cancer drug or inducer of an immune response has to find its way to those cancer stem cells. In addition, any *cancer stem cell*-targeted drug must not attack the normal stem cells of the body. One can imagine the complex cell–cell interactions between all of these different cell types, changing the normal gene expressions and phenotypes of those cells that are absent of those interactions. The apparent strategy is to design at least a multiple approach, first to kill the sensitive *cancer non-stem cells,* and then to eliminate at least some of the barriers to any *cancer stem cell*-targeted agent. However, remember that the death of those non-cancer stem cells will be releasing various cytokine-like chemicals that could cause the stimulation of the resistant and surviving cancer stem cells. Using stem cells grown in three-dimensions to mimic tissue organization, both tumor promoters and anti-cancer agents have been used to target the *cancer stem cells* [81,109].

## 8. Summary

In this *Commentary,* an examination of the origin of the *cancer stem cell* concept was developed by viewing historical experimental observations, techniques and concepts that have led to the operational concept that these *cancer stem cells* are those responsible for sustaining the growth of any tumor. Challenges to the understanding of what is the *cancer stem cell*, and from which cell it originated, have come from terms, such as *normal* cells, *immortal,* and *carcinogens*. These terms were examined in the context of several major classic concepts, such as the multi-stage, multi-mechanism process of carcinogenesis, and the demonstration of epigenetic agents as being the driver of this multi-stage, multi-mechanism process. A major challenge to the concept that *re-programming* of somatic differentiated cells exists during the *immortalization* of a normal cell. Characterization of several markers of isolated stem cells, such as Oct4A and connexin genes, in both normal adult stem cells and in *cancer stem cells*, has led to the potential strategy of targeting the cancer stem cells in a heterogeneous mixture of cancer cells in all tumors. Last, it has been speculated that there exist two different kinds of *cancer stem cells,* which suggests two very different kinds of anti-cancer drug strategy must be developed.

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
