# Peer review of "The Concept of “Cancer Stem Cells” in the Context of Classic Carcinogenesis Hypotheses and Experimental Findings"

_life, 2021, doi:10.3390/life11121308_

Round 1

Reviewer 1 Report

Reviewer Recommendation and Comments for Manuscript Number 1463913 submitted for publication in Life.

The Commentary entitled "The Concept of "Cancer Stem Cells in The Context of Classic Carcinogenic Hypotheses and Experimental Findings" by James E. Trosko is timely and exceptionally well written.

The author elaborates an exciting historical overview of the cancer stem cell concept, focusing on experimental findings and using his own experience of 50 years in the field.

Below are some minor typo corrections that need to be addressed to improve the article.

  • page 2, line 80
  • page 5, line 211, 222, 231
  • page 6, line 260

I also wonder what Professor Trosko thinks about the term "cancer stem-like cell" frequently used in the current literature.

Author Response

Reviewer #1,

a. Page 2 , line 80. Fixed

b. Page 5, lines 211, 222, 231. Fixed

c. Page 6, Line260. Fixed

d. What are my thoughts on "cancer stem cell like" cells. It would seem that "cancer stem cells" is a term that directly implies these cells behave similarly to normal cells in that they can only divide symmetrically to form two cancer stem cells or asymmetrically to form one daughter that maintains "cancer stem cell" property and the other to form a "cancer non-stem cell that is partially differentiated.

The "cancer stem cell-like" is a nuance that implies it is almost like a normal stem cell, in that it can divide symmetrically or asymmetrically, but the daughter cells of the "cancer-like stem cells are never normal, but either cancer stem cells or partially differentiated stem cells. I personally like "cancer stem cells", 

Reviewer 2 Report

This is an excellent review of Cancer Stem cells. The manuscript is appropriately referenced. It might be helpful to include a couple of summary figures. At least one tying together the authors view of gap junctions and Oct4 in cancer stem cells.

Author Response

First, thanks for the nice complement about this Commentary. While I would have loved to include a visual to link the stem cell characteristics and gap junctions, those figures are already in other published papers. I can only put a specific reference to a figure in a cited reference, which I did.

Reviewer 3 Report

This commentary develops interesting concepts on the definition of cancer stem cells and their interaction with the microenvironment. Also nice examples explaining the concepts are very well chosen. However, I suggest some improvements to make smoother the reading.

First of all, the title of the paragraphs need to be more focused. The first is an introduction, the second "Resolution of the origin of the “cancer stem cell” and the stem cell versus the de-differentiation hypotheses" is difficult to understand. Here there are two different concepts. One suggestion can be "The stem cell versus the de-differentiation hypotheses: the origin of the “cancer stem cell”". Why "cancer stem cell" is always kept in quotes? Also in line 65: "acquired" ? Instead of using many quotes ( also in the following text), the word can be underlined or written in italic.

In the third paragraph "Clues to resolving the stem cell verse De-differentiation hypotheses of the origin of cancer stem cells." I would change "resolving " with "prove" ( to correct also "verse" with "versus"). 

I suggest to delete the question markers from paragraph 4 and 5 because question markers do not really sound well for a commentary. 

In the title of paragraph 6 please delete "these" and use only "the"   “cancer stem cells”

I would move paragraph 6 to 7 and viceversa because I think it is more consequential to speak about the tissue specificity of cancer stem cells and later, make considerations on anti-cancer agents.

The general text should be re-read to delete some repetitions and mistakes. for instance: line 53-53 concern, concerning; 133-134: at that time; 147-148: it was shown; 151-152: population. line 364: please delete "remember", line 367: correct 3-dimesion with three-dimension. line 381-382: series. Line 442: "these" cancer stem cells, delete these and substitute with "the".

Lines 291-295: The phrase is not clear, please re-write.

Line 309-315: I am wandering if also senescence can be written as case of DNA damage. Aging is a physiological condition in which DNA modification occurs and can develop cancer.

Line 319: Please use the proper way to write gene names ( here "ras")

Line 437: I think it would be comprehensive to write 1 or 2 lines to briefly define what it is Precision Medicine ( again I suggest without quotes).

Author Response

Reviewer #3,

A. I have changed the Titles to the first 4 sections as reviewer suggested.

B. I have removed the quotation marks to "cancer stem cells: and used italics instead. I have still used the quotation marks on other important words for attention-grabbing.

C. I have changed "acquired" to a process by which drug resistance appears after treatment.

D. I have made changes in Title of Section 3 and also changed verse to versus, as well as changed prove from resolving.

E. Changed the question marks as suggested by Reviewer.

F. Changed "These" to " the" in Title of Section 6.

G. I liked the Reviewers suggestion to switch Section 7 to Section 6 and vice versa. This did create a massive change in the reference sequencing. However, I did made that change.

H. Made all suggested changes.

I. In response to this point: "I am wandering if also senescence can be written as case of DNA damage.", I would point out that DNA damage and mutations are not the "cause" of senescence. Reviewer is recommended to read this article: H.R. Choi, et al., " Restoration of senescent human diploid fibroblast by modulation of the extracellular matrix." Aging Cell 10: 148-157, 2011. I could add another page plus at least another 10 or so references to respond to your suggestion. However, it does not add to this Commentary.

J. Ras has been correct.

K. Added a definition of Precision Medicine.